# Growth Plate Skeletal Stem Cells and Their Actions Within the Stem Cell Niche

**DOI:** 10.3390/ijms26199460

**Published:** 2025-09-27

**Authors:** Natalie Kiat-amnuay Cheng, Shion Orikasa, Noriaki Ono

**Affiliations:** Department of Diagnostic and Biomedical Sciences, School of Dentistry, The University of Texas Health Science Center at Houston (UTHealth Houston), Houston, TX 77054, USAshion.orikasa@uth.tmc.edu (S.O.)

**Keywords:** growth plate, resting zone chondrocytes, skeletal stem cells, epiphyseal stem cell niche, parathyroid hormone-related protein (PTHrP), CD73, Axis inhibition protein 2 (Axin2), forkhead box protein A2 (FoxA2), apolipoprotein E (ApoE)

## Abstract

The growth plate is a specialized cartilage structure near the ends of long bones that orchestrates longitudinal bone growth during fetal and postnatal stages. Within this region reside a dynamic population of growth plate skeletal stem cells (gpSSCs), primarily located in the resting zone, which possess self-renewal and multilineage differentiation capacity. Recent advances in cell-lineage tracing, single-cell transcriptomics, and in vivo functional studies have revealed distinct subpopulations of gpSSCs, which are defined by markers such as parathyroid hormone-related protein (PTHrP), CD73, axis inhibition protein 2 (Axin2), forkhead box protein A2 (FoxA2), and apolipoprotein E (ApoE). These stem cells interact intricately with their niche, particularly after the formation of the secondary ossification center, through stage-specific regulatory mechanisms involving several key signaling pathways. This review summarizes the current understanding of gpSSC identity, behavior, and regulation, focusing on how these cells sustain growth plate function through adapting to biomechanical and molecular cues.

## 1. Introduction

Stem cells are defined by their dual capacity for self-renewal and multipotent differentiation [1,2]. These properties are shared across embryonic and induced pluripotent stem cells as well as tissue-resident populations such as hematopoietic and intestinal stem cells [1,3,4,5]. Skeletal stem cells (SSCs) represent a specialized subset within this broader hierarchy and are uniquely dedicated to the formation, maintenance, and repair of skeletal tissues [5,6,7,8]. Like other somatic stem cells, SSCs rely on supportive niches that regulate their balance between quiescence, proliferation, and differentiation [1,3]. However, in contrast to embryonic stem cells that generate diverse organ systems, SSCs are restricted to skeletal lineages, giving rise to chondrocytes, osteoblasts, osteocytes, and stromal progenitors [5,6,7,8]. Their localization is also distinct, with SSC pools identified in diverse skeletal tissues such as the resting zone of the growth plate, the periosteum, and the bone marrow. Thus, SSCs share the fundamental hallmarks of stemness with other stem cell types, but differ in their lineage restriction, anatomical localization, and functional specialization tailored to skeletal growth and regeneration.

The growth plate is a cartilage tissue located at the end of long bone and contains a specialized population of SSCs that direct longitudinal bone growth through precise cell division and maturation processes. These growth plate skeletal stem cells (gpSSCs) mainly reside in the resting zone, sustaining their stem cell properties through complex interactions with their surrounding microenvironment [9]. Studies have shown that these stem cells develop self-renewal capability and form columns of chondrocytes after the secondary ossification center (SOC) develops in the postnatal stage. The cell population demonstrates essential stem cell characteristics such as self-renewal capability and multipotency, with specific subpopulations identified by parathyroid hormone-related protein (PTHrP), CD73, axis inhibition protein 2 (Axin2), forkhead box protein A2 (FoxA2), and apolipoprotein E (ApoE) expression [6,7,10,11,12]. Through intricate molecular signaling networks and cellular interactions, these stem cells orchestrate skeletal development and tissue function throughout growth periods, controlled by key pathways including Wnt/β-catenin, mammalian target of rapamycin (mTOR), and Hedgehog signaling [6,7,10,13,14,15].

The secondary ossification center shapes the stem cell niche in the growth plate by delivering essential signals and mechanical cues that maintain epiphyseal stem cells [16]. This environment allows PTHrP^+^ chondrocytes to remain inactive within the resting zone until SOC formation, when they start to create stable monoclonal columns of chondrocytes through controlled asymmetric and symmetric cell divisions [6]. This microenvironment combines biochemical signals, mechanical forces, and cell–cell interactions to control stem cell behavior and tissue homeostasis (more details are discussed in the following Section 4. Key Signaling Pathway in Growth Plate Skeletal Stem Cells).

This review aims to provide a comprehensive overview of gpSSCs, highlighting their defining characteristics and functional roles in the postnatal stage. We further explore the dynamic cellular behaviors of gpSSCs within the growth plate and examine how these cells contribute to maintaining the function of the stem cell microenvironment, commonly referred to as the stem cell niche. Particular emphasis is placed on the intricate interactions between gpSSCs and their neighboring cell types and the regulatory signaling pathways that orchestrate their self-renewal, differentiation, and spatial organization. By integrating findings from recent lineage-tracing studies and single-cell transcriptomic analyses, this review seeks to advance our understanding of the complex biology of gpSSCs and their niche, thereby offering insights into potential therapeutic strategies targeting skeletal growth and regeneration.

## 2. Growth Plate Structure and Function

### 2.1. Characteristics of the Three Zones in Postnatal Growth Plate

The growth plate is a highly specialized cartilage structure in the epiphysis responsible for the longitudinal growth of long bones. Histologically, it is divided into three distinct zones: the resting zone, the proliferative zone, and the hypertrophic zone, arranged in that order from the epiphyseal to the metaphyseal side (Figure 1).

The resting zone constitutes the most epiphyseal portion of the growth plate and plays a foundational role in regulating endochondral ossification and longitudinal bone growth [17]. Although traditionally described as a morphologically quiescent region containing sparsely distributed, round chondrocytes embedded in abundant extracellular matrix, recent advances in cell-lineage tracing, transcriptomics, and single-cell analysis have redefined the resting zone as a dynamic and indispensable stem cell niche [6,7]. A defining characteristic of the resting zone is the presence of slow-cycling, self-renewing skeletal stem cells. Using a genetically inducible Cre-loxP system (*Pthrp-creER*), in vivo lineage tracing experiments have revealed that PTHrP^+^ cells originating from the resting zone contribute extensively to the proliferative and hypertrophic layers over time, particularly during postnatal growth [6]. Other studies have also supported the concept that the resting zone provides a robust source of all other chondrocytes in the growth plate, using multicolor in vivo clonal analysis [7], *Foxa2-creER* [11], and *Apoe-mCherry* [12] mice. These findings suggest that the resting zone functions as a reservoir of long-lived, multipotent stem cells.

The clonal architecture of resting chondrocytes also shows significant spatial organization. Individual clones in the resting zone expand radially, feeding into multiple proliferative columns and maintaining growth plate structure, and function over extended periods [7]. Beyond serving as a stem cell pool, the resting zone performs several critical physiological functions: (1) Regulation of Growth Plate Homeostasis: The resting zone integrates feedback signals, such as those from the PTHrP-Indian Hedgehog (Ihh) axis, to balance chondrocyte proliferation and differentiation. PTHrP produced by resting chondrocytes maintains the proliferative state as a forward signal and inhibits premature hypertrophic differentiation of adjacent chondrocytes [18,19,20,21,22,23]. (2) Mechanical Stabilization: Given its high ECM content and relatively low cellularity, the resting zone provides mechanical support to the overlying epiphysis and maintains the alignment of proliferative columns [24]. (3) Response to Injury and Regeneration: Under growth plate injury or disruption, resting chondrocytes have been shown to become activated, increase their proliferation, and contribute to tissue regeneration, underscoring their potential relevance in pediatric orthopedic repair strategies [11]. (4) Temporal Regulation of Growth: The resting zone also serves as a “growth timer.” As longitudinal growth decelerates during adolescence, the resting zone becomes progressively depleted of stem-like cells, leading to the eventual closure of the growth plate in humans [7,25].

Situated directly below the resting zone, the proliferative zone is composed of columnar chondrocytes aligned in parallel stacks oriented along the longitudinal axis of the bone. This zone is characterized by high mitotic activity, as chondrocytes in this region undergo rapid proliferation and directional expansion, contributing directly to longitudinal bone growth. The transition of resting chondrocytes into proliferative chondrocytes marks a key step in chondrocyte differentiation and spatial organization within the growth plate [26]. Proliferative chondrocytes are highly metabolically active and begin expressing matrix components and signaling molecules necessary for further maturation and hypertrophy. Functionally, proliferative chondrocytes act as a critical intermediary population that connects the quiescent, stem-like resting chondrocytes to the terminally differentiating hypertrophic chondrocytes. The orchestrated expression of transcription factors, growth factors, and matrix genes in this zone is essential for regulating growth plate expansion and endochondral ossification.

Located below the proliferative zone, the hypertrophic zone represents the terminal stage of chondrocyte differentiation within the growth plate. In this region, chondrocytes exit the cell cycle and dramatically increase cell volume—often up to a 10-fold enlargement—while simultaneously altering their gene expression profiles [27]. This hypertrophic transition is critical for initiating the subsequent steps of endochondral ossification, including cartilage matrix mineralization, vascular invasion, and recruitment of osteoprogenitor cells [18]. These cells no longer divide but actively secrete signaling molecules and enzymes that orchestrate matrix remodeling and ossification. Functionally, they act as a bridge between cartilage and bone. Interestingly, recent studies have demonstrated that a subset of hypertrophic chondrocytes can transdifferentiate into osteoblasts and marrow stromal cells, challenging the long-held view that these cells undergo terminal apoptosis [28,29]. This transdifferentiation, driven by a specialized gene expression program, is essential for coordinating the terminal steps of endochondral ossification and bone elongation.

### 2.2. Signaling Pathways Regulating the Growth Plate Activity

Growth plate chondrocytes are regulated by a multiple network of signaling pathways that interact to coordinate longitudinal bone growth. At the center of this regulation is the PTHrP–Ihh negative feedback loop, which maintains proliferative activity and prevents premature hypertrophy of chondrocytes, thereby preserving growth plate structure and function [18,19,20,21,22,23]. Hedgehog signaling, particularly Ihh, not only drives hypertrophic differentiation of chondrocytes but also induces osteoblast differentiation in adjacent perichondrial cells, acting both in PTHrP-dependent and -independent contexts [21,30,31]. Fibroblast growth factor (FGF) signaling exerts a critical role by regulating proliferation and differentiation through FGFR1 and FGFR3, with gain- or loss-of-function mutations leading to growth plate disorganization and skeletal dysplasia [32,33,34,35,36].

Endocrine signals such as the GH–IGF1 axis further contribute to growth plate regulation, as IGF1 supports chondrocyte proliferation and hypertrophy, and sustains the PTHrP–Ihh loop [37,38,39,40,41]. In parallel, Gsα-mediated cAMP–PKA signaling acts downstream of the PTHrP receptor to inhibit excessive Hedgehog activation, preventing accelerated hypertrophy and ectopic ossification [42,43,44,45]. Wnt/β-catenin signaling also interfaces with this network, promoting the transition of proliferating chondrocytes into hypertrophic chondrocytes and driving osteoblast differentiation, in part through direct interactions with Sox9 and Ihh [46,47,48,49]. At the transcriptional level, Runx2 functions as a master regulator of hypertrophic maturation and osteoblast commitment, partly by activating Ihh expression [50,51,52,53,54].

Moreover, BMP/TGF-β signaling is indispensable for chondrocyte proliferation and differentiation, and it enhances the PTHrP–Ihh regulatory mechanism to sustain growth plate activity [55,56,57,58,59,60,61,62,63,64]. Finally, recent work has highlighted the importance of mTORC1 signaling, which is required for maintaining the stem cell niche within the resting zone following secondary ossification [7,65]. Together, these interconnected pathways provide a robust molecular framework that ensures proper coordination of chondrocyte proliferation, hypertrophy, and lineage plasticity during postnatal skeletal growth.

## 3. Various Growth Plate Skeletal Stem Cells

### 3.1. PTHrP-Positive Skeletal Stem Cells in the Growth Plate

A comprehensive set of experimental approaches employing *Pthrp-mCherry* knock-in and *Pthrp-creER* transgenic mouse models was utilized to elucidate the identity, spatial–temporal localization, and stem cell properties of PTHrP-expressing cells within the postnatal growth plate [6]. Analysis of *Pthrp-mCherry* mice revealed that, during embryogenesis, PTHrP-mCherry^+^ cells were proliferative and localized to the Sox9^+^ perichondrial region. However, a striking shift in expression pattern was observed after birth. By postnatal day 3 (P3), a distinct population of non-proliferative PTHrP^+^ chondrocytes emerged centrally within the growth plate. Between P6 and P9, these cells expanded substantially to form a clearly demarcated resting zone. EdU incorporation assays demonstrated that these cells exhibited markedly reduced proliferative activity compared to proliferating zone chondrocytes, indicating a quiescent phenotype. To determine whether these PTHrP^+^ cells in the resting zone possess skeletal stem cell (SSC) properties, inducible lineage tracing experiments were performed using *Pthrp-creER* mice crossed with various *Rosa26* reporter strains (tdTomato, ZsGreen, and Confetti). Upon tamoxifen administration at P6, the fate of labeled PTHrP^CE^-P6 cells was traced over time. Initially retained within the resting zone, these cells began forming short columns, which progressively developed into long columns (>10 cells) spanning the proliferating and hypertrophic zones. Short columns peaked in number by P18 and declined thereafter, while long columns increased continuously through P36 (Figure 2A). Moreover, these labeled columnar chondrocytes remain for at least 1 year, suggesting that PTHrP^+^ resting chondrocytes serve as a long-term source of columnar chondrocytes, thereby fulfilling the defining criteria of self-renewing skeletal stem cells. In vivo clonal analysis using *Pthrp-creER; Rosa26-Confetti* mice further supported the notion that individual PTHrP^+^ resting chondrocytes give rise to entire columns in a clonal manner, as each column was uniformly labeled with a single fluorescent color. In addition, EdU label-retention experiments confirmed that these cells divide infrequently, consistent with their identity as slow-cycling, label-retaining stem cells. Long-term lineage tracing revealed that descendants of PTHrP^CE^-P6 cells contributed not only to the chondrocyte columns within the growth plate but also migrated beyond the hypertrophic zone to differentiate into Col1a1(2.3 kb)-GFP^+^ osteoblasts and Cxcl12-GFP^+^ marrow stromal cells in the metaphysis. These observations indicate that PTHrP^+^ cells possess multipotent differentiation potential in vivo, giving rise to both chondrogenic and osteo-stromal lineages. This study demonstrates that PTHrP-mCherry^+^ chondrocytes emerging in the postnatal resting zone constitute a unique population of skeletal stem cells with long-term self-renewal capacity and multilineage differentiation potential. The *Pthrp-mCherry* and *Pthrp-creER* mouse models together provide compelling evidence that PTHrP^+^ resting chondrocytes are essential for the longitudinal growth and homeostasis of the growth plate.

Recent work has elucidated the molecular mechanisms by which PTHrP-expressing chondrocytes in the resting zone of the postnatal growth plate acquire osteogenic potential [14]. Utilizing a tamoxifen-inducible *Pthrp-creER* mouse line crossed with *Ptch1-floxed* and *Rosa26-tdTomato* reporter alleles, the authors performed in vivo lineage-tracing and functional genetic experiments to specifically activate Hedgehog signaling in PTHrP^+^ resting chondrocytes. This genetic strategy allowed for the selective labeling and manipulation of a subset of slow-cycling chondrocytes residing exclusively in the resting zone, without affecting other skeletal cell populations. Following Hedgehog pathway activation via conditional deletion of Patched1 (Ptch1), PTHrP^+^ resting chondrocytes exhibited a marked loss of quiescence and underwent clonal expansion, forming concentrically arranged clusters termed “patched roses” within the resting zone (Figure 2B). These cells subsequently contributed to the formation of significantly widened columns of proliferating chondrocytes, leading to transient hyperplasia and distortion of the growth plate structure (Figure 2B). Long-term lineage-tracing studies revealed that a subset of these Hedgehog-activated PTHrP^+^ descendants migrated from the growth plate into the bone marrow cavity, where they differentiated into trabecular bone-forming osteoblasts. This osteogenic transformation was confirmed by the co-localization of tdTomato^+^ cells with Col1a1(2.3 kb)-GFP^+^ osteoblasts and osteoblastic markers such as osteopontin and type I collagen, as well as their incorporation into the bone matrix. Notably, similar Hedgehog activation in other chondrocyte populations (via *Col2a1-creER*, *Dlx5-creER*, or *Osx-creER*) failed to elicit comparable clonal expansion or osteogenic differentiation, underscoring the unique competence of PTHrP^+^ resting chondrocytes. These findings establish a novel paradigm in which Hedgehog signaling transiently confers clonal competency to PTHrP^+^ resting chondrocytes, facilitating their expansion and ultimate conversion into osteoblasts. This mechanism provides new insights into the regulation of skeletal stem cell fate decisions and highlights a potential target for enhancing bone formation during postnatal growth.

Another recent advance has shed light on PTHrP^+^ SSCs residing within the resting zone. That study provides compelling experimental evidence for the physiological and therapeutic relevance of these PTHrP-expressing stem cells [15]. This work not only affirms their critical role in longitudinal bone growth but also delineates the molecular mechanisms regulating their proliferation and activity. Through a combination of pharmacological and genetic approaches, the study demonstrated that activation of Hedgehog signaling significantly enhances the clonal expansion and proliferative activity of PTHrP^+^ SSCs. Genetic activation via conditional deletion of Ptch1 in PTHrP^+^ cells induced robust expansion of stem cell-derived clones and increased proliferative output across developmental stages, indicating an age-independent effect. In contrast, systemic pharmacological activation using Smoothened agonist (SAG) showed an age-dependent response, where early postnatal administration suppressed and later administration enhanced SSC activity. This discrepancy underscores the influence of the niche maturity and systemic context on the responsiveness of SSCs to Hedgehog stimulation. Interestingly, transcriptomic profiling of FACS-isolated PTHrP^+^CD73^+^ cells following SAG treatment revealed the emergence of a Wnt-inhibitory gene expression program, aligning with recent reports that a low-Wnt environment supports the maintenance of quiescent chondroprogenitors in the resting zone [13]. Thus, Hedgehog signaling not only directly promotes SSCs’ proliferation but also establishes a permissive microenvironment conducive to their expansion. The authors further explored the therapeutic potential of manipulating SSCs by locally implanting SAG-loaded beads into the SOC of rat femurs. This intervention led to a sustained increase in femoral and tibial length on the treated side, with no apparent adverse effects on joint cartilage, thereby offering proof-of-concept for correcting leg length discrepancies through local stem cell stimulation. These findings represent the first direct evidence that increasing the number of gpSSCs translates into enhanced linear bone growth in vivo.

### 3.2. CD73-Positive Skeletal Stem Cells in the Growth Plate

A previous study identified a novel population of SSCs residing in the growth plate, which acquires self-renewal capacity during postnatal development [7]. These cells were characterized by the surface expression of CD73 and were shown to possess both clonogenic potential and multipotency in vitro, differentiating into chondrocytes, osteoblasts, and adipocytes. Notably, under physiological conditions, these cells appear to function as unipotent stem cells restricted to the chondrogenic lineage. Using *Col2a1-creER; Rosa26-Confetti* mice, in vivo clonal analysis revealed a dramatic shift in the mode of clonal growth in the growth plate at approximately postnatal day 30. During fetal and early neonatal stages, growth plate columns were short and polyclonal, suggesting that multiple progenitors contribute to the columnar architecture. However, following SOC formation, long, monoclonal columns became dominant, indicative of sustained progeny derived from a single long-lived progenitor cell. This clonal transition marked the acquisition of self-renewal capacity by chondroprogenitors and the establishment of a functional stem cell niche. Immunohistochemical analysis revealed de novo expression of CD73 in the resting zone of the growth plate at P28, particularly adjacent to the SOC. Fluorescence-activated cell sorting (FACS) further demonstrated that CD73^+^CD49e^+^ chondroprogenitors expanded robustly in culture, formed significantly larger colonies than CD73^−^ cells, and exhibited tri-lineage differentiation potential in vitro. These findings substantiate their identity as adult stem cells. The spatial localization and behavior of CD73^+^ cells suggest that the resting zone, especially the region immediately beneath the SOC, functions as a stem cell niche. These progenitors undergo symmetric and asymmetric cell divisions, contributing to tissue homeostasis and columnar organization. Notably, the Hedgehog signaling pathway was shown to regulate the proliferation of these progenitors without affecting CD73 expression, while mTORC1 signaling modulated the balance between symmetric and asymmetric division. Perturbation of these signaling pathways led to disorganization of the resting zone or premature growth plate fusion, underscoring their critical role in maintaining the stem cell niche. Together, these findings provide compelling evidence that CD73^+^ chondroprogenitors in the resting zone of the postnatal growth plate represent a previously unrecognized population of skeletal stem cells. Their emergence is temporally and spatially associated with SOC formation and is essential for the continued supply of chondrocytes required for longitudinal bone growth.

### 3.3. Axin2-Positive Skeletal Stem Cells in the Growth Plate

A novel population of skeletal stem/progenitor cells marked by Axin2 expression, residing in the outermost layer of the postnatal mouse growth plate adjacent to Ranvier’s groove, was identified [10]. Utilizing *Axin2-creER; Rosa26-ZsGreen* mice, they performed lineage tracing following tamoxifen induction at postnatal day 6 and demonstrated that Axin2-positive cells initially localized to the periphery of the growth plate and expanded over time to contribute to the formation of the outer growth plate cartilage columns (Figure 3). These findings suggest that Axin2^+^ cells are responsible for the appositional (transverse) growth of the growth plate during early postnatal development. Immunohistochemical analyses revealed that these cells exhibit minimal or undetectable expression of Sox9 and Osterix, indicating a relatively undifferentiated state. Furthermore, EdU incorporation assays demonstrated that a subset of Axin2-positive cells is slow-cycling, a hallmark of stem cell populations. Longitudinal fate-mapping confirmed their sustained contribution to cartilage column formation in the outer growth plate, supporting their identity as skeletal stem cells with self-renewal and differentiation capacity. Functional evidence for their stemness was further strengthened by conditional ablation of β-catenin in Axin2^+^ cells. Loss of β-catenin resulted in impaired expansion of these cells toward the growth plate and the emergence of ectopic cartilaginous tissue within the Ranvier’s groove, highlighting the essential role of Wnt/β-catenin signaling in regulating both maintenance and lineage commitment of this progenitor population. Age-dependent analysis revealed a marked decline in the expansion and contribution of Axin2^+^ cells beyond six weeks of age, coinciding with the deceleration of longitudinal and transverse bone growth. Additionally, a sparse population of Axin2^+^ cells was observed within the resting zone of the growth plate, particularly during the growing phase, where they appeared to participate in the formation of new chondrocyte columns. Collectively, these findings establish Axin2^+^ cells as a distinct Wnt-responsive skeletal stem cell population that resides in the outermost layer of the growth plate. They exhibit low proliferative activity, slow-cycling characteristics, and are essential for appositional growth during postnatal development, acting in a spatiotemporally regulated manner under the control of Wnt/β-catenin signaling.

### 3.4. FoxA2-Positive Skeletal Stem Cells in the Growth Plate

FoxA2^+^ cells represent a distinct population of long-term skeletal stem cells (LT-SSCs) localized at the uppermost layer of the growth plate resting zone (RZ), specifically at the cartilage–bone interface adjacent to SOC (Figure 3) [11]. These cells are spatially and molecularly distinct from previously described PTHrP^+^ cells, which reside in the lower RZ. FoxA2^+^ cells are detectable as early as postnatal day 0 (P0), preceding the appearance of Col10^+^ hypertrophic chondrocytes around P7 in SOC. During early postnatal development, these FoxA2^+^ cells accumulate at the SOC–growth plate boundary and give rise to the top compartment of the RZ. Through in vivo lineage-tracing using *Foxa2-creER* mice, FoxA2^+^ cells were shown to persist within the RZ or differentiate into Col1^+^ osteoblasts within the SOC over time. Prior to P28, FoxA2^+^ cells exhibit dual osteo-chondrogenic potential, contributing to both the formation of the SOC and maintenance of the growth plate. After P28, their contribution to the SOC diminishes, and they become largely restricted to the growth plate cartilage. This temporal shift aligns with SOC maturation and indicates a context-dependent fate transition regulated by niche cues. Importantly, FoxA2 expression appears to mark epiphyseal cartilage prior to *Col10a1* expression, suggesting a role for FoxA2 as a pioneer factor during SOC formation. Functionally, FoxA2^+^ cells are characterized by their superior clonogenicity and longevity compared to PTHrP^+^ RZ cells. Colony-forming unit (CFU) assays demonstrated that FoxA2^+^ RZ cells form robust colonies, in contrast to hypertrophic zone-derived cells. While PTHrP^+^ cells exhausted their colony-forming capacity by passage five, FoxA2^+^ cells sustained this potential beyond passage nine. In serial transplantation assays, FoxA2^+^ (ZsGreen^+^) cells maintained their self-renewal and multipotency in vivo, giving rise to cartilage (Alcian blue staining), bone (Alizarin Red-S), and adipose (LipidTox) tissues even in tertiary recipients. These findings confirm that FoxA2^+^ cells fulfill key criteria of LT-SSCs. In a murine model of Salter–Harris type I (SH1)-like growth plate injury, FoxA2^+^ cells were shown to be critical for tissue regeneration. Following injury, the growth plate regenerated within seven days, restoring physeal cartilage without fibrocartilaginous or osseous scar formation. Lineage tracing revealed that FoxA2^+^ cells contributed to both Col10a1^+^ hypertrophic chondrocytes and Col1a1^+^ osteoblasts within the metaphyseal bone. Genetic ablation of FoxA2^+^ cells impaired cartilage regeneration, underscoring their essential role in post-injury repair. The activity and fate of FoxA2^+^ cells appear to be closely linked with SOC dynamics. Postnatally, FoxA2^+^ cells downregulate Ki67 expression after P5, coinciding with hypertrophic differentiation. The SOC may serve as a signaling hub to maintain FoxA2^+^ cell stemness, as previously suggested its role in maintaining chondroprogenitor renewal [7]. The transition from dual osteo-chondrogenic potential to chondrogenic restriction after P28 further supports a niche-dependent regulation of FoxA2^+^ LT-SSC fate. In conclusion, FoxA2^+^ LT-SSCs contribute to the formation and regeneration of the growth plate and SOC in a temporally and spatially coordinated manner. Their early dual-fate potential, prolonged self-renewal, and capacity for multilineage differentiation position them as a unique stem cell population vital for growth plate homeostasis. Moreover, their robust regenerative capacity highlights their potential in therapeutic strategies aimed at treating pediatric growth plate injuries and cartilage defects.

### 3.5. ApoE-Positive Skeletal Stem Cells in the Growth Plate

ApoE-positive cells represent a comprehensive population of resting chondrocytes in the growth plate and exhibit characteristics of skeletal stem cells. In a recent study, ApoE was identified as a novel and pan-marker for resting chondrocytes (RCs) in the growth plate (Figure 3) [12]. By combining single-cell RNA sequencing (scRNA-seq) of growth plate chondrocytes with spatial validation using RNA in situ hybridization (RNAscope), ApoE was found to be highly and specifically expressed in RC clusters, with over 97% of RCs expressing *ApoE.* Its expression was restricted to the resting zone and was absent in proliferative and hypertrophic zones. This spatially restricted expression was further confirmed in the human growth plate, indicating its cross-species conservation. To examine the in vivo characteristics of ApoE-expressing RCs, the authors generated a knock-in mouse model (*Apoe-mCherry*) in which mCherry expression reflects endogenous ApoE activity. mCherry^+^ chondrocytes emerged in the resting zone after the SOC formation and persisted with age, albeit in reduced numbers. An EdU pulse-chase experiment demonstrated that mCherry^+^ RCs are slow-cycling, supporting their identity as a quiescent cell population distinct from proliferating columnar chondrocytes. The stem/progenitor properties of ApoE^+^ RCs were further confirmed by in vitro assays. FACS was used to isolate CD45^−^CD31^−^Ter119^−^CD73^+^mCherry^+^ cells, effectively enriching ApoE^+^ RCs while excluding cells from bone marrow and vascular lineages. These isolated cells exhibited robust osteogenic and chondrogenic differentiation capacities, limited adipogenic potential, and significantly higher colony-forming ability than ApoE-negative cells—features consistent with skeletal progenitor/stem cell identity. Notably, the study also compared ApoE^+^ RCs with previously established RC stem cell markers such as *Pthrp*, *Axin2*, and *Foxa2*. Although most cells positive for these markers were contained within the ApoE^+^ population, only a small fraction of ApoE^+^ RCs expressed any of these genes. This observation highlights the heterogeneity of skeletal stem cells in the RZ and underscores the utility of ApoE as the first comprehensive marker that encompasses the entire RC population. Taken together, ApoE-positive cells reside specifically in the resting zone after SOC formation and demonstrate key features of skeletal stem cells, including slow proliferation, multipotency, and high self-renewal potential. The *Apoe-mCherry* reporter mouse provides a powerful tool to investigate the biology and functional heterogeneity of RCs and facilitates further exploration into the mechanisms regulating longitudinal bone growth.

## 4. Key Signaling Pathway in Growth Plate Skeletal Stem Cells

### 4.1. Stage-Specific Roles of Hedgehog Signaling in the Regulation of Growth Plate Skeletal Stem Cells

Several key signaling pathways intricately control gpSSCs’ function. Hedgehog signaling exerts stage-specific effects on the maintenance and activity of gpSSCs. Pharmacological modulation of this pathway before the formation of SOC markedly affects the gpSSC population [6]. Specifically, administration of either the Hedgehog pathway antagonist LDE225 or the agonist SAG at this early developmental stage led to a significant reduction in PTHrP-expressing SSCs and their progeny, highlighting the necessity of tightly regulated Hedgehog activity for SSC maintenance during early postnatal development. In contrast, Hedgehog signaling appears to assume a different regulatory role after SOC formation. A previous study reported that treatment with the Hedgehog antagonist vismodegib after SOC formation reduced the clonal expansion of SSCs and caused premature growth plate fusion around P35 [7]. Conversely, stimulation of Hedgehog signaling by SAG at the same stage promoted proliferation of resting zone chondrocytes, indicating a proliferative response in these cells to Hedgehog activation in a context-dependent manner.

### 4.2. mTORC1 as a Modulator of Skeletal Stem Cell Division and Resting Zone Structure in the Growth Plate

The mTORC1 signaling pathway plays a crucial role in regulating the cellular behavior of skeletal stem cells within the epiphyseal growth plate. Although activation of mTORC1 in chondrocytes does not significantly alter their proliferation or differentiation, it profoundly influences the organization and function of the growth plate stem cell niche (Figure 3). Specifically, conditional ablation of TSC1, a negative regulator of mTORC1, in Col2a1-expressing chondrocytes leads to hyperactivation of mTORC1 signaling and results in a progressive disorganization of the resting zone after SOC formation [7]. In these mice, clonal analysis revealed an increased number and size of multi-columnar clones, suggesting that mTORC1 activation shifts the balance toward symmetric division of stem cells. This expansion of colony-forming gpSSCs occurred without an accompanying increase in their proliferation rate, indicating a specific effect on stem cell fate rather than cell cycle regulation. Immunohistochemical analysis further showed enhanced expression of the stem cell marker CD73 within the disorganized resting zone of Tsc1-deficient mice, supporting the expansion of the gpSSC population. Importantly, these gpSSCs retained responsiveness to Hedgehog signaling, and pharmacological inhibition of the pathway induced their premature differentiation into columnar chondrocytes. This suggests that mTORC1 primarily modulates the mode of division and spatial organization of gpSSCs, while Hedgehog signaling governs their proliferative capacity and maintenance. In conclusion, mTORC1 signaling is a key regulator of growth plate stem cell dynamics, influencing the balance between self-renewal and differentiation through control of symmetric versus asymmetric cell division. Disruption of this balance may contribute to pathological conditions affecting growth plate function and longitudinal bone growth.

### 4.3. Wnt-IRX Axis in the Regulation of Growth Plate Skeletal Stem Cell Fate

Recent studies have revealed that hypertrophic chondrocytes (HCs) within the growth plate possess multipotent differentiation potential, contributing to osteoblasts and adipocytes, thereby positioning them as a key downstream population of skeletal stem cells. In this context, the previous study provides critical insights into the molecular mechanisms regulating lineage commitment of HC-derived skeletal progenitors, with a particular focus on the transcription factors IRX3 and IRX5 of the Iroquois homeobox family [66].

IRX3 and IRX5 were robustly expressed in late-stage HCs and in osteoblast lineage cells, including those in the trabecular and cortical compartments of endochondral bone. Functional ablation of either Irx3 or Irx5 in mice resulted in marked osteopenia and a concomitant increase in bone marrow adiposity. Notably, conditional deletion of Irx3 in HCs on an Irx5-null background led to a significant reduction in HC-derived osteoblasts and an increase in HC-derived adipocytes, suggesting that IRX3 and IRX5 cooperatively direct HC fate toward osteogenesis while suppressing adipogenic differentiation. Further mechanistic investigation demonstrated that IRX3 and IRX5 act downstream of canonical Wnt/β-catenin signaling. HC-specific β-catenin gain and loss-of-function models revealed that Irx3 and Irx5 expression are positively regulated by β-catenin activity. In vitro luciferase reporter assays confirmed that Wnt3a stimulation enhanced the promoter activity of both Irx3 and Irx5, supporting the view that these transcription factors are direct transcriptional targets of Wnt signaling in hypertrophic chondrocytes. These findings establish IRX3 and IRX5 as critical molecular switches that operate downstream of Wnt/β-catenin signaling to promote osteoblast differentiation and inhibit adipocyte lineage commitment in HC-derived skeletal progenitors. This regulatory axis not only elucidates a previously unrecognized mechanism underlying growth plate skeletal stem cell differentiation but also provides a conceptual framework for understanding the balance between osteogenesis and adipogenesis in bone development and homeostasis.

## 5. Conclusions

Growth plate skeletal stem cells represent a critical component of postnatal bone development, contributing not only to longitudinal growth but also to the maintenance and regeneration of the epiphyseal cartilage. These stem cells, residing predominantly in the resting zone, display remarkable heterogeneity and plasticity, as reflected by their distinct molecular markers and responses to signaling pathways (Figure 3). Recent studies have clarified that Hedgehog signaling exerts stage-specific effects on gpSSC proliferation and maintenance, mTORC1 controls the mode of stem cell division and the organization of the resting zone, and the Wnt-IRX axis directs lineage allocation toward osteogenesis while suppressing adipogenesis. Together, these pathways form an interconnected regulatory network that underpins the function of the growth plate stem cell niche. While most insights summarized in this review are derived from mouse models, it is important to note that several stem cell markers identified in mice do not always translate directly to humans. For instance, markers such as CD73 or FoxA2, which define mouse gpSSC subpopulations, have not yet been validated in human growth plates. However, ApoE has recently been identified as a pan-marker of resting chondrocytes in both mouse and human growth plates [12], suggesting partial conservation across species. Future studies are required to establish the extent to which the cellular hierarchies and signaling pathways described in mice apply to the human growth plate. Further investigations will be essential to determine the extent to which the gpSSCs populations and signaling pathways defined in murine models are conserved in the human growth plate. Elucidating these mechanisms will not only deepen our understanding of skeletal stem cell biology but also provide a conceptual framework for developing therapeutic approaches to growth plate injuries and skeletal dysplasia.

## Figures and Tables

**Figure 1 ijms-26-09460-f001:**
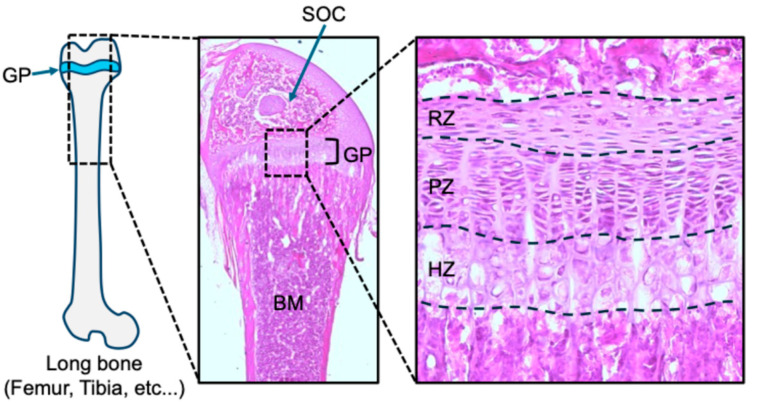
Morphology of femur and growth plate at postnatal day 21 from C57BL/6 mouse. A distal femur and magnified growth plate with hematoxylin and eosin staining. GP: growth plate, SOC: secondary ossification center, BM: bone marrow, RZ; resting zone, PZ; proliferative zone, HZ: hypertrophic zone.

**Figure 2 ijms-26-09460-f002:**
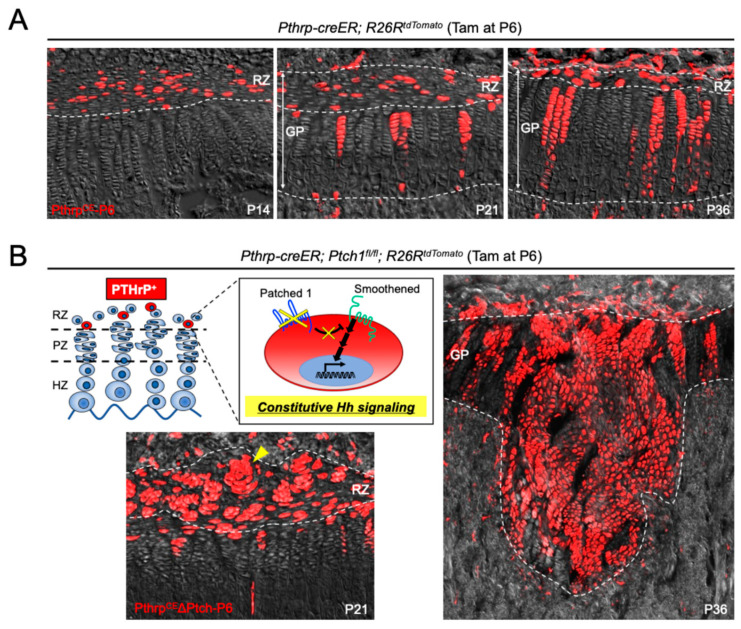
PTHrP^+^ resting zone chondrocytes are gpSSCs contributing to columnar formation, and Hedgehog signaling activates their osteogenic cell fate. (**A**) In vivo lineage-tracing of PTHrP^+^ resting zone chondrocytes using *Pthrp-creER; R26R^tdTomato^* bigenic mice injected with tamoxifen at P6 and chased up to P36. (**B**) Constitutive Hedgehog activation in PTHrP^+^ resting zone chondrocytes using *Pthrp-creER; Ptch1^fl/fl^; R26R^tdTomato^* trigenic mice injected with tamoxifen at P6 and chased up to P36. RZ: resting zone, GP: growth plate, Red: tdTomato, Gray: DIC, Arrowheads: patched rose. Adapted from Orikasa et al., *JCI Insight*. 2024;9(2):e165619 [14]. CC-BY-4.0.

**Figure 3 ijms-26-09460-f003:**
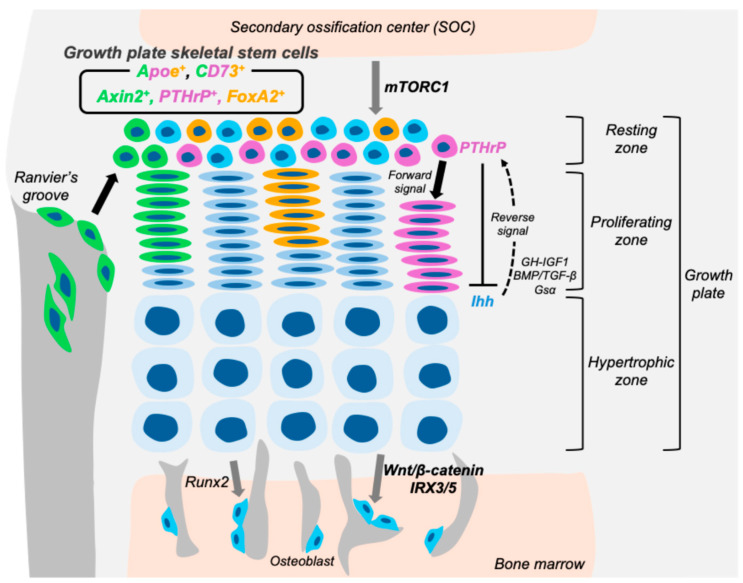
Growth plate skeletal stem cells and their actions within the stem cell niche. A diagram of growth plate skeletal stem cells and their actions within the stem cell niche. PTHrP, CD73, FoxA2, or ApoE-positive cells are present within the growth plate resting zone and serve as progenitors that give rise to downstream growth plate chondrocytes and osteoblasts. In contrast, Axin2-positive cells originate from the Ranvier’s groove and merge into the growth plate, where they contribute to the lateral expansion of the growth plate and provide downstream chondrocytes or osteoblasts. The formation of the secondary ossification center facilitates the development of a stem cell niche within the postnatal epiphyseal plate, a process sustained by mTORC1 signaling. Additionally, IRX3 and IRX5 function as key downstream effectors of Wnt/β-catenin signaling, promoting osteogenic differentiation while suppressing adipogenic commitment in hypertrophic chondrocyte-derived skeletal progenitors.

## Data Availability

No new data were created or analyzed in this study. Data sharing is not applicable to this article.

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
