# Peer review of "Growth Plate Skeletal Stem Cells and Their Actions Within the Stem Cell Niche"

_ijms, 2025, doi:10.3390/ijms26199460_

Round 1
Reviewer 1 Report
Comments and Suggestions for Authors
This review article was a nice summary of the currently known cell lineages and signaling mechanisms regulating the formation and maintenance of the growth plate and its niche. This review seems to be restricted to mouse models. Since we know several skeletal stem cell makers identified in mouse models do not translate to humans, my biggest question was, how do these cell lineages and signaling mechanisms translate to humans. A brief statement addressing this would suffice.
Also, “premature growth plate fusion” is mentioned several times in the manuscript, but not when this occurs. Please provide details of when this typically occurs in mouse models so there is a reference to how loss of specific signaling mechanisms or cell lineages disrupts growth plate formation and maintenance.
Line 28 pgSSCs should be defined here since it’s the first time it is mentioned in the main manuscript.
Lines 173-175 “These findings suggest that PTHrP+ resting chondrocytes serve as a long-term source of columnar chondrocytes, thereby fulfilling the defining criteria of self-renewing skeletal stem cells.” This seems like an overinterpretation of the data presented since cells were traced from P6 to P36. A brief statement of how Prhrp-CreER labeling a P6 is maintained for at least a year would help support this claim.
Author Response
Response to Reviewer A
Thank you very much for your constructive comments and critiques.
- <Reviewer>
This review article was a nice summary of the currently known cell lineages and signaling mechanisms regulating the formation and maintenance of the growth plate and its niche. This review seems to be restricted to mouse models. Since we know several skeletal stem cell makers identified in mouse models do not translate to humans, my biggest question was, how do these cell lineages and signaling mechanisms translate to humans. A brief statement addressing this would suffice.
<Response>
We appreciate the reviewer’s important point. To address this, we have added a statement in the conclusion noting the limitations of mouse models and emphasizing the need for further work on human growth plates. We also highlight ApoE as a recently validated cross-species marker.
The following sentence was added to “5. Conclusion” (Line 475-488):
While most insights summarized in this review are derived from mouse models, it is important to note that several stem cell markers identified in mice do not always translate directly to humans. For instance, markers such as CD73 or FoxA2, which define mouse gpSSC subpopulations, have not yet been validated in human growth plates. However, ApoE has recently been identified as a pan-marker of resting chondrocytes in both mouse and human growth plates [6], suggesting partial conservation across species. Future studies are required to establish the extent to which the cellular hierarchies and signaling pathways described in mice apply to the human growth plate. Further investigation will be essential to determine the extent to which the gpSSCs populations and signaling pathways defined in murine models are conserved in the human growth plate. Elucidating these mechanisms will not only deepen our understanding of skeletal stem cell biology but also provide a conceptual framework for developing therapeutic approaches to growth plate injuries and skeletal dysplasia.
- <Reviewer>
Also, “premature growth plate fusion” is mentioned several times in the manuscript, but not when this occurs. Please provide details of when this typically occurs in mouse models so there is a reference to how loss of specific signaling mechanisms or cell lineages disrupts growth plate formation and maintenance.
<Response>
We thank the reviewer for this suggestion. We have now specified the approximate timing (P30–P35, corresponding to ~4–6 weeks of age in mice) when premature growth plate fusion occurs in relevant sections.
The following sentence was modified in “4-1. Stage-Specific Roles of Hedgehog Signaling in the Regulation of Growth Plate Skeletal Stem Cells” (Line 407-410):
In contrast, Hedgehog signaling appears to assume a different regulatory role after SOC formation. A previous study reported that treatment with the Hedgehog antagonist vismodegib after SOC formation reduced the clonal expansion of SSCs and caused premature growth plate fusion around P35 [7].
- <Reviewer>
Line 28 gpSSCs should be defined here since it’s the first time it is mentioned in the main manuscript.
<Response>
We agree with the reviewer and have now defined gpSSCs at their first mention in the main text.
The following sentence was modified in “1. Introduction” (Line 42-44):
These growth plate skeletal stem cells (gpSSCs) mainly reside in the resting zone, sustaining their stem cell properties through complex interactions with their surrounding microenvironment [9].
- <Reviewer>
Lines 173-175 “These findings suggest that PTHrP+ resting chondrocytes serve as a long-term source of columnar chondrocytes, thereby fulfilling the defining criteria of self-renewing skeletal stem cells.” This seems like an overinterpretation of the data presented since cells were traced from P6 to P36. A brief statement of how Prhrp-CreER labeling a P6 is maintained for at least a year would help support this claim.
<Response>
We appreciate this valuable comment. We have added the new text to clarify that Pthrp-CreER labelled cell at P6 is maintained up to at least one year, thereby supporting their long-term stem cell identity.
The following sentence was modified in “3-1. PTHrP-Positive Skeletal Stem Cells in the Growth Plate” (Line 187-190):
Moreover, these labelled columnar chondrocytes remain in growth plate for at least 1 year, suggesting that PTHrP⁺ resting chondrocytes serve as a long-term source of columnar chondrocytes, thereby fulfilling the defining criteria of self-renewing skeletal stem cells.
Reviewer 2 Report
Comments and Suggestions for Authors
The authors Cheng, Orikasa, and Ono present a review manuscript
focusing on "cartilage structure near the ends of long bones" termed
the 'growth plate'. The topic is in line with the special issue topic
and the authors try to give a comprehensive overview on the field.
The research area is of interest to science, particularly to shed light on
the differences in the systems biology of organ stem cells
compared to differentiated cells of the tissue.
Unfortunately, this aspect is not really worked out in the given
manuscript.
The literature is surprisingly sparse with 23 publications.
I am not an dedicated expert here, but it seems to me that
the recent general stem cell literature might provide some more analogies
which could be taken up, at least as a hypothesis for the presented field,
or switching the perspective, some cross-links to other stem cell
research fields might make the overview on this specific field
more comprehensive and conclusive and less isolated.
Overall, the entire structure of the review is conceptually split
in several sections which are not nicely building a whole picture.
This might be resulting from the chosen section headers
(not following a concept) but also from that, that the three authors
put their work together, without adjusting the work entities
to a logical story.
At least my concern is, that this is more an enumeration of elements
than a true review.
Maybe a resubmission could address all the mentioned aspects below
which are by far not complete.
Further comments
Abstract
Well written -
Introduction
26ff
For non experts a better definition of the tissue locations of the
'growth plate', 'resting zone', 'secondary ossification center',
'microenvironment', 'stem cell niche' etc. would be necessary -
so Figure 1 should come in early here not at the end of the manuscript.
At the end of the manuscript a second figure needs to be placed
with some mechanistic pathways (systems biology) summing up
the present regulatory situation
and the difference of the typical patterns defining these
stem cell and the differentiated cell which in many cases
is a different expression profile of a panel of molecules.
Would be nice to see such a graph also here, maybe based on
TCGA expression database.
To discuss Figure 1 already here:
The background of the Figure 1 should also give a physiological image,
a top view of the bone/cartilage area and
some histology cross-sections through a bone/cartilage area where
the 'growth plate', 'resting zone' is localized.
So additionally to the scheme also two more panels.
BTW, the scheme itself needs improvement.
43-44
Needs to be more detailed - otherwise it is only name dropping.
45ff
The authors set their objectives on a very high level.
Realistically, the authors should pick up some core objectives
which are also reflected by figures showing the advances
in their field.
81
The four citations should be assigned to certain aspects which
should be explained in a more detailed fashion.
84
Citation scheme switched from number to name.
87-100
This section is only creating a scheme, which is ok. But this scheme
needs to be embedded into the published literature which is not done,
except at the last line -
Is the scheme adopted from these two publications? - and how are
these two publications underlining each of the mentioned aspects?
Are they coherently supportive? or do they own contradictions?
...
Overall the section 2.1 needs a revision. It contains several
redundancies, not all of which are really necessary.
The section needs to have a much more stringent structure.
Furthermore, the citation scheme is not fine-grained.
135ff
Chapter 2.2. : The title needs some abstraction: Mechanistic elements
in the growth plate -- or something like that...
Now one pathways is given -- is that all what is known?
Or is this only the newest aspect?
A figure to this section is advised.
155ff
3.1.
A revision is advised & the creation of a figure which might inspire
the author to improve the stringency of the text which is
without a good structure / guiding idea:
a summary figure is missing.
244ff
3.2.
245
Newton again, with a wrong citation scheme.
A summary figure is missing.
276ff
3.3.
277
Wrong citation scheme: Usami et al. (2019)
A summary figure is missing.
307 to 449 ditto.
451ff
The conclusion is not perfectly summarizing the wealth
of information given before, especially the complex pathway
situation.
As mentioned above, a new summarizing figure would be
advised here, because Figure 1 is better placed at the beginning
of the manuscript.
Author Response
- <Reviewer>
Abstract - Well written
<Response>
We thank the reviewer for this positive evaluation.
- <Reviewer>
Introduction
26ff For non experts a better definition of the tissue locations of the 'growth plate', 'resting zone', 'secondary ossification center', 'microenvironment', 'stem cell niche' etc. would be necessary - so Figure 1 should come in early here not at the end of the manuscript.
To discuss Figure 1 already here: The background of the Figure 1 should also give a physiological image, a top view of the bone/cartilage area and some histology cross-sections through a bone/cartilage area where the 'growth plate', 'resting zone' is localized. So additionally to the scheme also two more panels. BTW, the scheme itself needs improvement.
<Response>
We thank the reviewer for pointing out the need for clarity for non-expert readers. We have created a new Figure 1, an anatomical illustration of the growth plate using an actual tissue section (cited in Line 78). Additionally, previous Figure 1 has been revised and reorganized into Figure 3 to summarize the growth plate skeletal stem cell populations and signaling pathways within the growth plate.
- <Reviewer>
43-44 Needs to be more detailed - otherwise it is only name dropping.
<Response>
We appreciate the reviewer’s comment. We have now clarified that more details are described in the following section, in Section 4 (Line 60-61).
- <Reviewer>
45ff The authors set their objectives on a very high level. Realistically, the authors should pick up some core objectives which are also reflected by figures showing the advances in their field.
<Response>
We appreciate the reviewer’s suggestion. We have now appended two figures and tied these with the main text to improve readability and conceptual streamlining.
- <Reviewer>
81 The four citations should be assigned to certain aspects which should be explained in a more detailed fashion.
<Response>
We appreciate the reviewer’s suggestion. We have now clarified the sentence by assigning a specific citation to a defined aspect of the topic.
The following sentence was modified in “2. Growth Plate Structure and Function” (Line 86-91):
Using an inducible Cre-loxP system (Pthrp-creER), in vivo lineage tracing experiments have revealed that PTHrP+ cells originating from the resting zone contribute extensively to the proliferative and hypertrophic layers over time, particularly during postnatal growth [6]. Other studies have also supported the concept that the resting zone provides a robust source of all other chondrocytes in the growth plate, using multicolor in vivo clonal analysis [7], Foxa2-creER [11], and Apoe-mCherry [12] mice.
- <Reviewer>
84 Citation scheme switched from number to name.
244ff 3.2. 245 Newton again, with a wrong citation scheme.
276ff 3.3. 277 Wrong citation scheme: Usami et al. (2019).
307 to 449 ditto.
<Response>
We appreciate the reviewer’s careful attention to detail. We have revised the citation scheme to ensure consistency.
The following sentences were modified:
(Line 95-97) Individual clones in the resting zone expand radially, feeding into multiple proliferative columns and maintaining growth plate structure, and function over extended periods [7].
(Line 169-172) A comprehensive set of experimental approaches employing Pthrp-mCherry knock-in and Pthrp-creER transgenic mouse models was utilized to elucidate the identity, spatial-temporal localization, and stem cell properties of PTHrP-expressing cells within the postnatal growth plate [6].
(Line 206-208) Recent work has elucidated the molecular mechanisms by which PTHrP-expressing chondrocytes in the resting zone of the postnatal growth plate acquire osteogenic potential [14].
(Line 234-235) That study provides compelling experimental evidence for the physiological and therapeutic relevance of these PTHrP-expressing stem cells [15].
(Line 260-261) A previous study identified a novel population of SSCs residing in the growth plate, which acquires self-renewal capacity during postnatal development [7].
(Line 292-294) A novel population of skeletal stem/progenitor cells marked by Axin2 expression, residing in the outermost layer of the postnatal mouse growth plate adjacent to Ranvier's groove was identified [10].
(Line 354-355) The SOC may serve as a signaling hub to maintain FoxA2⁺ cell stemness, as previously suggested its role in maintaining chondroprogenitor renewal [7].
(Line 366-368) In a recent study, ApoE was identified as a novel and pan-marker for resting chondrocytes (RCs) in the growth plate (Figure 3) [12].
(Line 442-445) In this context, the previous study provides critical insights into the molecular mechanisms regulating lineage commitment of HC-derived skeletal progenitors, with a particular focus on the transcription factors IRX3 and IRX5 of the Iroquois homeobox family [66].
- <Reviewer>
87-100 This section is only creating a scheme, which is ok. But this scheme needs to be embedded into the published literature which is not done, except at the last line - Is the scheme adopted from these two publications? - and how are these two publications underlining each of the mentioned aspects? Are they coherently supportive? or do they own contradictions?
<Response>
We appreciate the reviewer’s thoughtful comments. Following the reviewer’s comments, we have added the published literature to the second and third clauses.
The following citations were added:
(Line 102-105) 2) Mechanical Stabilization: Given its high ECM content and relatively low cellularity, the resting zone provides mechanical support to the overlying epiphysis and maintains the alignment of proliferative columns [24].
(Line 105-108) 3) Response to Injury and Regeneration: Under growth plate injury or disruption, resting chondrocytes have been shown to become activated, increase their proliferation, and contribute to tissue regeneration, underscoring their potential relevance in pediatric orthopedic repair strategies [11].
- <Reviewer>
Overall the section 2.1 needs a revision. It contains several redundancies, not all of which are really necessary. The section needs to have a much more stringent structure. Furthermore, the citation scheme is not fine-grained.
<Response>
We appreciate the reviewer’s careful attention to detail. We revised the section to remove redundancies and correct the citation scheme. This significantly improved the accuracy and readability of the section.
The following sentences were removed in “2-1. Characteristics of the Three Zones in Postnatal Growth Plate”:
- Among these, the resting zone, located adjacent to the epiphysis, was historically considered a quiescent region with low cellular turnover [3]. However, studies have revealed that this zone harbors a population of stem-like cells that play a central role in maintaining growth plate homeostasis and contributing to endochondral ossification over time [11].
- Resting chondrocytes within this zone are small, round cells lacking the columnar alignment seen in more distal zones. These cells are characterized by low nuclear-cytoplasmic ratios, dense chromatin (i.e., heterochromatin), and minimal organelle development—features consistent with a low metabolic and proliferative state. The extracellular matrix in this region is rich in type II collagen and proteoglycans, contributing to a dense and resilient structure that anchors the proliferative columnar chondrocytes below.
- Proliferative chondrocytes exhibit a more differentiated phenotype than their quiescent counterparts in the resting zone. They form characteristic vertical columns, reflecting both clonal expansion and the organization of daughter cells along the direction of growth.
- Hypertrophic chondrocytes are morphologically distinct, displaying an enlarged cytoplasm, condensed chromatin, and reduced organelle density.
- This transdifferentiation process underscores the emerging concept of lineage plasticity within the growth plate and its contribution to bone formation. In summary, hypertrophic chondrocytes represent the final differentiation stage of growth plate chondrocytes.
The following sentences were modified:
(Line 79-81) The resting zone constitutes the most epiphyseal portion of the growth plate and plays a foundational role in regulating endochondral ossification and longitudinal bone growth [17].
(Line 95-97) Individual clones in the resting zone expand radially, feeding into multiple proliferative columns and maintaining growth plate structure, and function over extended periods [7].
(Line 118-120) Proliferative chondrocytes are highly metabolically active and begin expressing matrix components and signaling molecules necessary for further maturation and hypertrophy.
(Line 132-133) Functionally, they act as a bridge between cartilage and bone.
(Line 136-138) This transdifferentiation, driven by a specialized gene expression program, is essential for coordinating the terminal steps of endochondral ossification and bone elongation.
- <Reviewer>
At the end of the manuscript a second figure needs to be placed with some mechanistic pathways (systems biology) summing up the present regulatory situation and the difference of the typical patterns defining these stem cell and the differentiated cell which in many cases is a different expression profile of a panel of molecules. Would be nice to see such a graph also here, maybe based on TCGA expression database.
135ff Chapter 2.2. : The title needs some abstraction: Mechanistic elements in the growth plate -- or something like that... Now one pathways is given -- is that all what is known? Or is this only the newest aspect?
A figure to this section is advised.
<Response>
We thank the reviewer for this excellent suggestion. Unfortunately, cancers/tumors of the bone, cartilage or growth plate (osteosarcoma and chondrosarcoma) are not available in the TCGA database. To address the reviewer’s point, we retitled the section as “Signaling Pathways Regulating the Growth Plate Activity”, expanded the content to include Hedgehog, FGF, GH-IGF1, cAMP-PKA, Wnt/β-catenin, BMP/TGFβ and mTORC1 signaling, modified Figure 3, and added references. We believe that this modification has strengthen the mechanistic aspect of the regulation of the growth plate activity.
The whole sentences in section 2-2 were modified to the following (Line 139-166):
Growth plate chondrocytes are regulated by a multiple network of signaling pathways that interact to coordinate longitudinal bone growth. At the center of this regulation is the PTHrP–Ihh negative feedback loop, which maintains proliferative activity and prevents premature hypertrophy of chondrocytes, thereby preserving growth plate structure and function [18-23]. Hedgehog signaling, particularly Ihh, not only drives hypertrophic differentiation of chondrocytes but also induces osteoblast differentiation in adjacent perichondrial cells, acting both in PTHrP-dependent and -independent contexts [21,30,31]. Fibroblast growth factor (FGF) signaling exerts a critical role by regulating proliferation and differentiation through FGFR1 and FGFR3, with gain- or loss-of-function mutations leading to growth plate disorganization and skeletal dysplasia [32-36].
Endocrine signals such as the GH–IGF1 axis further contribute to growth plate regulation, as IGF1 supports chondrocyte proliferation and hypertrophy, and sustains the PTHrP–Ihh loop [37–41]. In parallel, Gsα-mediated cAMP–PKA signaling acts downstream of the PTHrP receptor to inhibit excessive Hedgehog activation, preventing accelerated hypertrophy and ectopic ossification [42–45]. Wnt/β-catenin signaling also interfaces with this network, promoting the transition of proliferating chondrocytes into hypertrophic chondrocytes and driving osteoblast differentiation, in part through direct interactions with Sox9 and Ihh [46–49]. At the transcriptional level, Runx2 functions as a master regulator of hypertrophic maturation and osteoblast commitment, partly by activating Ihh expression [50–54].
Moreover, BMP/TGFβ signaling is indispensable for chondrocyte proliferation and differentiation, and it enhances the PTHrP–Ihh regulatory mechanism to sustain growth plate activity [55–64]. Finally, recent work has highlighted the importance of mTORC1 signaling, which is required for maintaining the stem cell niche within the resting zone following secondary ossification [7,65]. Together, these interconnected pathways provide a robust molecular framework that ensures proper coordination of chondrocyte proliferation, hypertrophy, and lineage plasticity during postnatal skeletal growth.
- <Reviewer>
155ff 3.1. A revision is advised & the creation of a figure which might inspire the author to improve the stringency of the text which is without a good structure / guiding idea: a summary figure is missing.
244ff 3.2. 245 A summary figure is missing.
276ff 3.3. 277 A summary figure is missing.
307 to 449 ditto.
<Response>
We are grateful for this careful observation. For Section 3-1, we created a new Figure 2A,B (referenced in Line 187, 216, 219) based on our in vivo lineage tracing data. For the other sections, we have improved the previous Figure 1 and moved it to Figure 3 (referenced in Line 297, 324, 368, 420, 470). This diagram reflects the localization of each skeletal stem cell population within the growth plate, the direction of cell migration and proliferation, and the signaling pathways within the growth plate. We believe that these changes facilitate the understanding of the concept.
- <Reviewer>
451ff The conclusion is not perfectly summarizing the wealth of information given before, especially the complex pathway situation.
<Response>
We thank the reviewer for this valuable comment regarding the conclusion section. We improved the conclusion to capture the information presented in the manuscript, particularly regarding the complex signaling pathways that regulate growth plate skeletal stem cells (gpSSCs).
The following sentence was added to “5. Conclusion” (Line 470-475):
Recent studies have clarified that Hedgehog signaling exerts stage-specific effects on gpSSC proliferation and maintenance, mTORC1 controls the mode of stem cell division and the organization of the resting zone, and the Wnt-IRX axis directs lineage allocation toward osteogenesis while suppressing adipogenesis. Together, these pathways form an interconnected regulatory network that underpins the function of the growth plate stem cell niche.
- <Reviewer>
As mentioned above, a new summarizing figure would be advised here, because Figure 1 is better placed at the beginning of the manuscript.
<Response>
We thank the reviewer for this excellent suggestion. We have created a new Figure 1 (Line 78), an anatomical illustration of the growth plate using an actual tissue section, as requested by the reviewer. Additionally, previous Figure 1 has been revised and reorganized into Figure 3 to summarize the growth plate skeletal stem cell populations and signaling pathways within the growth plate.